# Investigation of Factors Related to Sport-Specific Compulsory Element Execution in Artistic Swimming

**DOI:** 10.3390/sports12040096

**Published:** 2024-03-30

**Authors:** Vivien Laski, Dóra Ureczky, Márta Wilhelm

**Affiliations:** 1Doctoral School of Biology and Sport Biology, Faculty of Sciences, University of Pécs, 7622 Pécs, Hungary; laskivivi@gmail.com; 2Institute of Sport Sciences and Physical Education, Faculty of Sciences, University of Pécs, 7622 Pécs, Hungary; udora21@gmail.com

**Keywords:** sports performance, artistic swimming, force plate test, hand grip strength

## Abstract

Artistic swimming is an Olympic sport requiring a high level of fitness, as well as technical skills, artistry, flexibility, a good sense of rhythm, remarkable lung capacity and physical strength. The artistic swimming of adolescents has been largely untouched by the scientific community, so training this group based on scientific data is difficult. Due to the gap between theoretical knowledge and practical application, this study aimed to measure the technical elements, strength parameters, and swimming performance of young artistic swimmers and to compare swimming performance and strength with the quality of artistic element performances. Hungarian female junior artistic swimmers (14.36 ± 1.01 year) participated in this study. Swimming tests were performed, and three basic elements (body boost, barracuda, and vertical position) were scored. A negative correlation was found between 100 m freestyle swimming times and vertical position scores, as well as between 50 m breaststroke leg swimming times and body boost scores. Moreover, three months of breaststroke leg swim training resulted in improvements in body boost performance. In addition, a positive correlation was found between mean hand grip strength and barracuda scores, as well as between vertical position scores. According to the results of three months of breaststroke leg training, swimming practice improves artistic swimming performance by itself. A correlation was found between strength parameters and the execution of synchro elements, showing the importance of complex training strategies in this sport.

## 1. Introduction

Artistic swimming (AS) is an Olympic sport requiring a high level of fitness, technical skills and artistry. Athletes must possess dynamism like gymnasts [1,2], flexibility like rhythmic gymnasts, a good sense of rhythm like dancers [1], increased lung capacity like divers [3,4] and physical strength like swimmers [5]. In addition, AS requires a high level of coordination and combines the skills of aerobic and anaerobic fitness (endurance, speed and flexibility) with the ability to learn performance and choreography quickly [6]. As each AS element has a defined movement execution, athletes strive to accomplish perfect arm and legwork techniques. Sculling with the hands and upturns or tucks with the legs are very important in AS elements. AS athletes are required not only to maintain body positions for long durations but also to move, drive, rotate and change directions in the water [7,8]. Muscle strength, core stability and motor control are essential for the efficient execution of technical moves and the initiation of required functional limb movements [9,10]. Proper legwork is equally indispensable, and its significance has been studied in detail in kinematic research [11]. Opening the hip joint in a sidewise manner involves keeping the knees wider during the eggbeater kick [11], while for arm movements, not only a symmetry [12] matters, but also arm strength and muscle mass are important. Moreover, the height achieved by an athlete performing different elements is influenced by the horizontal position of the body out of the water. Hence, it is important to maintain a balanced and stretched body in a vertical position (VP) while performing the sculling movement of the upper limbs [12].

Although many studies have been conducted in different areas of AS (the technique of one element [11,12], body composition [13,14,15]) and morphofunctional indicators [16], several other areas require additional exploration. Because of this expanded scope, we widened the range of our research. It would be worth investigating whether there is an ideal appearance for artistic swimmers. Aesthetics are very important, but body composition might influence performance. To achieve adequate performance and avoid various eating disorders, it is important to optimize protein and fat intake, as well as body composition [14]. No previous study has been found examining the body composition of young artistic swimmers.

AS is a technical sport, similar to swimming, in which a precisely defined sequence of movements must be performed. The characteristic elements of AS are body jumps, wherein athletes jump out of the water. Two important thrust techniques are barracuda (BC) and body boost (BB) [6,17]. Athletes must perform these thrusts to achieve the highest VP, and as they are both force movements, they require rapid and maximal force to perform them effectively [18]. BC and BB are considered power moves in artistic swimming. After analysis, high reliability was found for both the sport-specific tests and the countermovement jump (CMJ) [6].

Although the nature of swimming is different from AS, in both cases, athletes adapt to the special conditions provided by the aquatic environment [18,19]. Although flat and support sculling in artistic swimming have been examined in previous kinematic studies [12,20], no article has been found that shows similarities between the arm and leg positions performed in AS and swimming (S). In both sports, optimal angles of arm and leg motion are required during forward and upward movements for energy conservation. The necessity of the study would be reinforced by focusing on the influence of swimming training on AS performance [21]. In both freestyle swimming and VP, the arms move at a 90-degree angle to the propulsive, pulling section. Furthermore, VP, like swimming, requires a cyclic motion and defined vertical lane (with the body in one line) throughout the movement. In BC, the athlete reaches VP in an instant, breaking out of the pre-position (submerged pike). There are several types of arm techniques in BC (e.g., the flip, rotate, and press technique and the crossover technique). In practice, the position and dynamics of the trunk are as important as the technique used to execute the movement. Some studies have compared the physique or performance [5,19] of swimmers with artistic swimmers, but none have reported the beneficial effects of swimming as a complementary and preparatory sport for AS. Hand grip strength is very important in various sports, including AS. Athletes use hand power for lifting, pitching, throwing, and maintaining a specific position, as well as to improve their performance and technical development. Hand grip force determines the performance of various artistic swimming elements (e.g., VP, ballet leg, split position, spins and BC) by using hand strength [22] and forearm muscles [23]. In competitive sports, different types of training may influence, maintain, and improve the performance of athletes [1]. In AS, upper and lower limb strength was examined as a key factor in performing compulsory elements (such as explosiveness in the case of breakout elements, i.e., BB and BC) [17]. Success in this sport is based on the execution of compulsory elements. Understanding the protocol and mode of training that improve AS element performance is still incomplete and requires further investigation. In case of detailed research, both sport performance and injury prevention would be improved. 

The age group of 12- to 15-year-olds is unexplored in the field of AS research, despite being significant in talent selection [24]. Athletes at this age have already participated in many years of training in AS; therefore, it is necessary to study the adolescence age group as wide and complex as possible, keeping in mind that in the given time frame, athletes develop rapidly, learning artistic swimming skills, and, with appropriate training methods, these processes might be accelerated. Applying additional swimming into training routines to improve the performance of AS elements and promote strength development might help coaches in their efforts. During this period, adolescents experience significant physical changes, including growth spurts, changes in body composition, and development of strength and flexibility [25,26]. These changes have a profound impact on their abilities to perform technical elements and routines in artistic swimming. Adolescents in the 12–15 age group are in a critical phase of skill acquisition and refinement. They have the cognitive and motor skills necessary to learn complex movements and sequences, making it an ideal time to develop and master the fundamental skills of artistic swimming [26].

No study investigating elements of artistic swimming in this age group at the competitive level has been published yet. Data might help reinforce the importance of swim training (especially breaststroke) in this population.

After analyzing data in the background literature, we hypothesized that (i) body composition would influence AS performance, and (ii) swimming performance (SP) would affect the quality of AS element performances. In the current research, our aims were as follows: (a) to determine parameters for identifying changes in body composition during one training season that modify AS performance; (b) to compare swimming times with the execution of artistic elements; and (c) to measure hand grip and lower limb strength in comparison with AS element execution. In our study, we also aimed (d) to examine the relationship between AS performance and swimming time and to correlate jump height with these measurements.

## 2. Materials and Methods

### 2.1. Experimental Approach

In this study, young artistic swimmers were examined to determine whether athletes demonstrating better SPs also achieved better AS scores. The age of the athletes was between 12 and 15 years. There are not many AS athletes in Hungary (approx. 2500); therefore, it was difficult to find participants for the measurements. To collect the highest amount of data, several tests were conducted in approximately the same group of subjects. All participants have been practicing AS for at least four years and all of them were competitors. The training protocol of the athletes was the same throughout the last year.

The exercise protocol was developed by our group based on literature data. To ensure the sustainability of this research, it was necessary to categorize athletes according to their level of ability. Therefore, subjects were classified using a six-stage scale derived from Alana’s study in 2022, which assessed the spectrum of training backgrounds and athletic abilities [27]. In this study, each participant belonged to the Tier 3 group, i.e., they were highly trained and competing at the Hungarian national level.

In addition, athletes were examined to determine whether strength parameters affected AS performance. Body composition and performance tests were conducted in different AS groups. As a novelty in the literature, competitive levels were also measured [28]. In addition, regression models were used for the valid and reliable description of certain factors affecting competitive performances [29].

### 2.2. Participants

A total of 17 artistic swimmers (Tier 3: highly trained/national level) were tested in this study (*n* = 17), and their body compositions were monitored for one season. The first measurement (T1) was performed in March 2017 and the last test (T2) in December 2017. The mean age of the athletes was 14.36 ± 1.01 years at the beginning of this study and 15.1 ± 1.02 years at the end, years spent in training were 6.90 ± 2.18. The athletes attended four training sessions per week, each lasting 3 h, and consisting of 1 h of land training and 2 h of practice in water. The following variables were measured in the same population of AS athletes: hand grip strength (HGS), BC, and VP. HGS data were compared with the execution heights of BC and VP.

Participants and their parents were thoroughly informed of the measurement procedures. This study was approved by the Ethics Committee of the University of Pécs (license number: PTE6749, 7 June 2017), and all parents provided consent for their participation. The Institute of Sport Sciences and Physical Education of the University of Pécs also approved this protocol. The tests adhered to the Code of Ethics of the World Medical Association.

The selection criteria were as follows: participants were required to be within a specific age range, with an average age of approximately 14 years at the time of the study. They were elite artistic swimmers classified at Tier 4 (international level), indicating a high level of proficiency and competitive experience in the sport. Participants were required to have a minimum number of years of training in artistic swimming, with an average of 5.4 years of training experience. All participants were active competitors in the Hungarian Artistic Swimming Association, ensuring their active involvement in competitive AS. Athletes were members of rural artistic swimming teams located in southern Transdanubia and central Transdanubia, providing a specific geographical context for the study. The study included only female participants.

The exclusion criteria were as follows: participants with medical conditions, such as cardiovascular or respiratory diseases, musculoskeletal injuries, or any other condition that could pose a risk during physical activity or affect their ability to perform the required movements were excluded. Individuals with recent injuries, especially those affecting the upper or lower extremities, spine, or joints crucial for artistic swimming movements, were also excluded from the study.

### 2.3. Body Composition

Body composition was measured with Inbody720 (Biospace, Korea). The measured parameters were weight (W), body fat mass (BFM), right-hand mass (RHM), left-hand mass (LHM), skeletal muscle mass (SMM), right-leg mass (RLM), left-leg mass (LLM) and the total protein amount of the body. Body composition measurements were conducted at the beginning (T1) and the end (T2) of the season. At the time of the first body composition measurements, athletes performed swimming tests (100 m freestyle and 200 m medley) and lower limb strength tests (force plate tests were conducted with Tenzi Ltd., Pilisvörösvár, Hungary). The participants were members of one AS team in the southern Transdanubia region of Hungary. Prior to conducting any measurements, participants were instructed to abstain from eating or drinking for several hours (usually 4–6 h). They were also asked to empty their bladder and remove any excess clothing or accessories that could interfere with the measurements. Body composition measurements were always taken at the same time of the day.

### 2.4. Examination of Artistic Swimming Elements

Force and explosive force are very important factors in AS analysis, due to their applicability and predictability in determining the performance of AS elements [2]. Swimming times were compared with the performance of AS elements.

The evaluation of standard elements was conducted after a standard swimming/AS warm-up in each case. Each element was performed in the same place in the swimming pool (one meter from the edge of the pool and above the black track sign below the water’s surface). The measured elements were performed in a 50 m long, 2 m deep pool (water temperature: 27 ± 1 °C) and were scored independently by three official experienced judges (each judge held a valid judge license), and the FINA (Fédération internationale de natation) International Points System was used [30].

Members of the studied group performed BB, BC, and VP. Each element was performed three times, and the best result was used and analyzed. The judging panel scored the athletes in competition conditions. Judges scored each element (0–10-point scale) independently, and the calculated average score was used. The judges were positioned one meter away from the edge of the pool, and the performing athlete was positioned in front of 3 judges. The scores of AS elements were compared with the swimming performance.

Vertical position: This is a strength exercise in which an athlete must keep their legs elevated for 10 sculls while the head, hips, and ankles are in one line. The athlete’s body is inverted (head down) and balanced, supporting the sculling movement with the upper limbs.

Barracuda thrust: The starting position is a submerged pike, and the legs are perpendicular to the surface of the water. The athlete performs a rapid vertical upward movement, and, at the same time, the body unrolls to a VP. The head, hips, arms, and ankles should be in one line at the end of the movement. Maximum height is desirable while performing this movement.

Body boost: This is a movement using the leg muscles to propel the body (without touching the bottom of the pool) straight up out of the water. The head and the hips should be in a vertical line. The body boost movement starts with the aid of the upper limbs sculling in the water and then swinging above the head.

### 2.5. Dry-Land Measurements

#### 2.5.1. Hand Grip Strength Test

Upper limb strength was tested using hand grip strength. Each athlete held the measuring device (a digital hand dynamometer, Camry, model: SCACAM- EH101, Sensun Weighing Apparatus Group Ltd, Guangdong, China) in a standing position with a lowered arm clamping the handle of the dynamometer with full force (measured in kg). All athletes were right-handed. The athletes performed the task with both hands and repeated it three times, with a one-minute resting period between repetitions. The average of the three scores was calculated. For future reference, these exercises were called the right-hand grip strength (RHG, dominant hand), left-hand grip strength (LHG), and two hands mean (hand grip strength mean (HGS)) tests. The first measurement was performed in March 2017 and the second in December 2017.

#### 2.5.2. Force Plate

Lower limb strength was measured while a subject performed the maximum possible vertical jump, and the vertical force exerted on the ground was measured using a force platform (Tenzi Kft. Pilisvörösvár, Hungary) at a frequency of 420 Hz. CMJ was performed in the following way: athletes were initially in a standing position on the power platform and performed a jump with both legs and arms swung to the maximum possible height. They performed 3 jumps; the best one was used for further analysis. All participants were measured on the same day, after a predetermined warm-up. There were no other tests performed that day.

### 2.6. Swimming Performance Test

In addition, the following SP tests were conducted: 50 m of breaststroke swimming using only legs and 100 m of freestyle. Breaststroke leg swimming times were measured again three months later. SP was measured at the same time in a group using a (SPARTAN, model: JS-510, Fuzhou Lexinda Electronic Co., Fuzhou, China) stopwatch. We investigated the effect of breaststroke leg swim training in this study, since there was no previous research examining this topic. The effect of breaststroke leg swim training on artistic performance (Results Section 3.2.1) was tested in 38 elite artistic swimmers (Tier 4: international level) (age: 14.47 ± 1.69 years, training years 5.4 ± 2.7 years)**,** and the correlation between AS performance and breaststroke leg swimming times was analyzed (all participants were competitors in the Hungarian Synchronized Swimming Association). We examined if there was a significant improvement in BB performance between the two test times. Tested athletes were members of one rural AS team, and the average time spent in training was 5.4 ± 2.7 years. The measured performance variables were body boost (BB) and breaststroke leg swimming test scores. In addition, the relationship between breaststroke leg swimming (swimming without using arms) and BB was analyzed. Athletes examined were members of three rural AS teams in southern Transdanubia and central Transdanubia (20 females, their age at the first test was 13.09 ± 0.73 years). Based on the result of a one-time measurement, a 3-month-long breaststroke leg swim training program (3 times/week, each session lasting 1 h) was carried out in addition to their regular training protocols, and the effects of the swim training was tested in a follow-up study. The control group (13.05 ± 0.74 years) undertook general swim training at the same time, 3 times a week, with each session lasting for 1 h. No differences were found in the ages of the athletes (*p* = 0.23). The initial BB performance (March 2017) was compared with the second test conducted three months later (May). Training conditions were the same: swimming was performed in one lane, one by one, and AS elements were performed in two or more lanes (at a 2 m water depth and water temperature of 27 ± 1 °C).

### 2.7. Statistical Analysis

The Statistica 10.0 program was used for the statistical analysis. Means and standard deviations were calculated for the analyzed variables and tested for normality (with the Lilliefors test based on the Kolmogorov–Smirnov test). Continuous variables were compared using either a one-sample Student’s *t*-test or the Wilcoxon test, as appropriate. Bivariate analysis was performed using Pearson’s and Spearman’s rank correlation analyses. Differences and associations were considered significant at *p* < 0.05. A regression equation was calculated between the strength tests and the AS elements (with a general regression model and simple regression).

## 3. Results

### 3.1. Study

#### 3.1.1. Body Composition and Artistic Swimming Elements

Body composition measurements were conducted twice, at the beginning (T1) and the end (T2) of the season (*n* = 17). Some significant differences were found between the time points. The relevant data are shown in Table 1.

Significant changes were found between the two time points (T1 and T2) in weight (*p* = 0.007), protein amount (*p* = 0.02), skeletal muscle mass (*p* = 0.03), right-leg muscle mass (*p* = 0.002), and left-leg muscle mass (*p* = 0.002). Significant changes in body composition are marked with asterisks (*) (*p* < 0.05). No significant differences were found in body fat mass—*p* = 0.74, right-hand muscle mass—*p* = 0.13, and left-hand muscle mass—*p* = 0.18.

The artistic swimming elements were scored at the beginning of the season, as mentioned in the Section 2. The average scores were 7.45 ± 0.39 for BB; 7.2 ± 0.35 for BC; and 5.72 ± 0.72 for VP. No correlation was found between body composition data and AS performance (Table 2).

No significant correlation was found between body composition and artistic swimming performance.

#### 3.1.2. Hand Grip Strength

The AS element (BC and VP) scores correlated with the athletes’ hand grip strengths. The correlations are presented in Table 3.

A correlation was found between both hand grip strength and barracuda performance. Vertical position scores significantly correlated with the right (stronger) hand strength only. Asterisks (*) represent statistical significance, * = *p* < 0.05.

A significant difference was found between the RHG mean (26.52 ± 4.36) and LHG (24.29 ± 3.36; *p* = 0.001). The right hand of all subjects was dominant (all athletes were right-handed), and it was found to be significantly stronger than the left hand. Correlation of VP was found with left-hand strength (r = 0.61), but no correlation was found with right-hand strength (*r* = 0.47). The HGS mean correlated with BC (r = 0.62) and VP (r = 0.59) scores. A regression model was created, wherein RHG correlated with BC scores, and LHG correlated with VP and BC scores. A linear regression line was drawn between HGS and the AS elements, VP = 1.19 × LHG + 4.7 and VP = 0.7 × HGS + 4.1. The regression model was composed of BC and HGS, as follows: BC = 3.76 × RHG + 4.3, BC = 2.79 × LHG + 2.95, and BC = 3.61 × HGS + 2.7. It was mainly the dominant hand that exerted a higher force.

#### 3.1.3. Force Plate Measurements

The mean force (CMJ) of the group was 7.35 ± 0.87 Ns/kg. Lower limb strength was correlated with the AS element scores, and a correlation was found between the CMJ mean and BB performance scores (7.46 ± 0.34; r = 0.61; *p* = 0.01). The regression model was composed of CMJ and BB, as follows: BB = CMJ × 0.329 + 5.04.

#### 3.1.4. Swimming and Artistic Swimming Performance Tests

The effect of SP on AS element scores was analyzed by measuring 100 m freestyle swimming performance (89.53 ± 5.54 s). There was a negative correlation between swimming times (89.53 ± 5.5 s) and VP scores (5.72 ± 0.72; r = −0.72; *p* = 0.001), but a correlation was also found between freestyle swimming times and BC scores (7.21 ± 0.37; r = −0.53; *p* = 0.19).

### 3.2. The Effect of Swim Training on Artistic Performance

#### 3.2.1. Swimming Performance

The AS athletes’ (*n* = 38) 50 m breaststroke leg swimming time was 60.74 ± 5.7 s. The swimming times of subjects were negatively correlated with BB scores (7.35 ± 0.98) (r = −0.58, *p* = 0.0001), with the athletes swimming faster obtaining higher BB scores.

#### 3.2.2. Three-Months-Long Breaststroke Leg Swim Training

After three months of breaststroke leg swim training (*n* = 20), swimming times were measured as 62.93 ± 7.11 s in February and 61.8± 6.59 s in May. Significant improvements were not found in the breaststroke leg swimming times (*p* = 0.82), while BB scores improved significantly (Z = 3.81; *p*-value = 0.0001). The average scores were 6.84 ± 0.96 in February and 7.53 ± 0.82 in May. No significant difference was found between the BB performance in February (6.76 ± 0.85) and May (6.79 ± 0.9) *p* = 0.47, and between the 50 m breaststroke leg swimming times in February (62.6 ± 6.2 s) and May (62.6 ± 6.2 s) (*p* = 0.82), as measured in the control group (*n* = 19).

## 4. Discussion

Healthy ways to perform physical exercise for children are currently gaining increased attention. Many children become acquainted with modern sports, such as AS. Adapting to this sport is complex due its special and difficult training system (gymnastics, dance, and swimming skills are all required), and athletes reaching higher levels of competitive performance are rare.

This study aimed to measure the technical elements, strength parameters, and SP of young artistic swimmers and compare SP and physical strength with the quality of artistic element performances. Additional swim training (breaststroke leg) was conducted to measure its effect on the execution of technical scales. It has been shown that different morphological factors (height, fat mass, shoulder strength, and upper extremity length) affect SP [29,31,32,33,34]; nevertheless, no significant correlation was found between body composition and AS performance in this study, similarly to the findings of other studies [2,13,35]. Unfortunately, only few have studied these factors in AS athletes [13,15]. However, during a yearlong study, significant differences were found in some body composition parameters (e.g., W, SMM, RLM, and LLM), but BFM, LHM, and RHM did not change notably, while significant changes in mass (SMM, LLG, and RLG) indicate the importance of these muscles. Moreover, a correlation was found between strength and AS element performances. These results emphasize that muscle strength is an important component of success in this sport. The relationship between VP scores and HGS underlines that performing VP can be considered a strength exercise, wherein the athlete must maintain a VP for a long time while performing various leg exercises. The correlation between LGS and VP scores show that the weaker arm is very important because it determines how the VP is executed. To improve the execution of a VP, the performance of the weaker (left) arm should be developed to match that of the stronger (right) arm. Since VP is a symmetrical pose, the strength of the weaker/left arm is more important. This study demonstrated (similar to [22]) that arm strength plays an important role in this sport, as shown in the correlation between the AS element scores and HGS. Grip strength positively correlated with faster sprint swimming performance in both master and elite level swimmers, and improved grip strength resulted in faster sprint freestyle swimming [36]. Similar to upper limb strength, lower limb also plays a crucial role in AS performance. The correlation between BB scores and vertical jump performance suggests that the strength of the lower limbs is important in the correct performance of BB and BC [18,37]. Similar to our results, Morris et al. (2019) found that the strength of limbs are key factors in swimming [37]. The relationship between BB and BC performance is noteworthy (r = 0.66), since the two elements are executed in opposite directions but with the same dynamics. The relationship between the two AS elements underlines the importance of explosive power in this sport, since both tasks require fast and dynamic execution. Moreover, both movements are performed with a vertical body posture, explosive force, and the coordinated work of the trunk muscles. Athletes need strong muscles that function properly, as neuromuscular connections are crucial for performing AS elements as accurately as possible [10,30].

The results of this study demonstrate the relationship between SP and the AS element scores of young artistic swimmers. For both types of swimming (freestyle and breaststroke leg), the faster the athletes completed the swimming distance, the higher the scores they achieved in performing mandatory elements, suggesting that the exerted force and water perception greatly influence the performance of AS elements. The correlation between swimming times and VP scores indicates that an athlete’s position at the moment of touching the water plays an important role in both SP and AS, since they use the same technique.

Although legwork did not improve significantly during a 3-month-long breaststroke training period, those who were faster in breaststroke leg swim tests achieved significantly higher BB scores. Therefore, BB performance improved remarkably after breaststroke leg swim training, while there was no BB improvement in the control group. The breaststroke leg swim training likely contributed to the technical development of BB, strengthening the lower limb muscles, since a correlation was found between swimming and BB performance. These data may help to reinforce the importance of strength and swimming training (especially breaststroke) in this population. Successful performance in AS depends on swimming abilities for the execution of the AS routine of elaborate movements in water [38,39].

In this sport, arm and leg movements are very important for perfecting performance, since technical knowledge improves the execution of compulsory elements. All three elements require the coordinated work of the trunk, leg, and arm muscles to execute their final position in a straight vertical line.

This study has some limitations. Differences in the developmental motor coordination skills of adolescents were not considered. However, the studies were mainly conducted in adolescents and growing subjects. The follow-up study was not sufficient for monitoring the fully developed stages of adolescence. Moreover, this study did not consider the menstrual cycles of girls despite the fact that hormonal changes modify the performance of athletes. As with body changes during maturation, these are important performance factors. Furthermore, most of the studies included in this research focused on adults. Therefore, we recommend conducting further tests in both growing subjects and adults to define AS development guidelines. To address these limitations and further advance in understanding the effects of AS training, future research should contemplate several directions. Firstly, studies with larger and more diverse samples of artistic swimmers could provide more evidence and enhance the generalizability of the findings. Longitudinal studies tracking swimmers over an extended period could also elucidate the long-term effects of training on performance, skill development, and injury prevention. Additionally, future research could explore the impact of individual differences, such as age, gender, and skill level, on the effectiveness of training programs. Comparative studies in different training methods and interventions could also shed light on the most effective approaches for enhancing performance and skill acquisition in AS.

## 5. Practical Applications

Our results are in accordance with other studies, showing that one side (usually the right) of the body is stronger, and the body of a swimmer might float unequally on the surface of water. This fact should receive more attention, since these inequalities could be balanced with adequate training, thus decreasing the possibility of injury or suboptimal performance. As such, we recommend that coaches focus on strengthening athletes’ non-dominant sides. Furthermore, the relationship between anthropometric data, body composition, and AS elements should be studied in more depth. The mean age of measured athletes was 15.1 ± 1.02 years (at the end of the test), and body contours are more feminine at this age. In addition, body ratios change due to puberty; hence, body fat and hip ratios increase. It would be important to examine the same parameters in young adult female athletes whose bodies are fully developed, and body ratios will not change.

## 6. Conclusions

In our study, the importance of strength and swimming skills of young female AS athletes was demonstrated. A correlation was found with the execution of AS elements in both force tests. Regardless of the differences between dominant and non-dominant hand force, the relationship between HGS and the performance of AS elements was strong. Comparing data obtained from other athletes in different sports with those of artistic swimmers would help the generalization of our findings. Swimming is a highly technical sport, and training methods to improve overall performance have been explored both on land and in water. Could practicing AS elements also improve SP due to the similarities between the two sports? This is an important question, since kinesthesia is a key factor in both sports [10,21,40]. If so, this may provide a new opportunity for swimmers to improve their techniques. Based on research, it could be implemented based on the kinetics of movement in water, such as velocity and technique, which are linked to the energy cost of swimming [41,42]. AS performance is associated with velocity variability, similar to elite backstroke swimmers who are able to control it through several adaptations to overcome the high drag and inertia [43]. Even though breaststroke leg swim training did not improve the BB scores, it might play a major role in mastering the technical execution of BB. Many coaches still underestimate the importance of the technical aspects, even though swimming technique contributes the most to AS performance [40]. Other factors that could have influenced the performance in this sport and the outcomes of this study include the swimmers’ experience and the amount of time spent in the water. The level of sensitivity and skills developed over years of practice might serve as a complementary factor to the more physical aspects of this sport, and the observed results may not be as pronounced when considering a more experienced sample.

## 7. Perspective

The results of this study demonstrate the relationships between SP and AS element scores. Although it has been shown that different physiological factors (height, fat mass, strength, and upper extremity length) affect SP [18,22,29,33,34,40], very few studies have investigated similar factors in AS athletes [2,13].

Collecting, analyzing, and correlating the performance data of AS athletes, such as strength, flexibility, and body composition, could help trainers and practitioners to create personalized training sessions for young athletes, according to age groups. This approach may help avoid or reduce the risk of injuries or fallouts in water sports. The results of this study, along with morphofunctional indicators [16] and definitions of the competitive levels [27], may help to complete the picture of success in AS.

Providing specific recommendations or guidelines for trainers and practitioners based on our findings can enhance its practical relevance and applicability. Some recommendations resulting from the aforementioned points are as follows: Considering the age groups of young athletes, trainers can adjust the intensity, duration, and complexity of training sessions accordingly. Based on the performance data of AS athletes, trainers and practitioners can design individualized training programs tailored to the specific needs and capabilities of each athlete. Younger athletes may require more emphasis on skill development and technique refinement, while older ones may focus on building strength and endurance. For example, athletes with lower strength levels may benefit from targeted strength training exercises, while those with limited flexibility could focus on improving mobility and stretching routines.

It would be worthwhile examining the anthropometric data and body compositions of adult athletes in this sport.

## Figures and Tables

**Table 1 sports-12-00096-t001:** Body composition data of artistic swimmers at the beginning (*n* = 17; 14.36 ± 1.01 years) and end of the season.

	T1 Mean	T2 Mean
Weight (kg)	54.27 ± 6.77	55.5 ± 6.82 *
Body fat mass (kg)	11.97 ± 4.28	12.19 ± 3.90
Right-hand mass (kg)	2.02 ± 0.26	2.06 ± 0.31
Left-hand mass (kg)	2.01 ± 0.25	2.06 ± 0.31
Protein (kg)	8.31 ± 0.78	8.51 ± 0.90 *
Skeletal muscle mass (kg)	23.08 ± 2.34	23.65 ± 2.78 *
Right-leg mass (kg)	6.44 ± 0.74	6.64 ± 0.85 *
Left-leg mass (kg)	6.39 ± 0.75	6.59 ± 0.87 *

**Table 2 sports-12-00096-t002:** Correlation data of body composition and artistic swimming performance (*n* = 17).

	Barracuda	Vertical Position	Body Boost
Weight	r = 0.32	r = 0.26	r = 0.84
Body fat mass	r = 0.29	r = 0.28	r = 0.87
Skeletal muscle mass	r = 0.23	r = 0.18	r = 0.69
Right-hand mass	r = 0.31	r = 0.45	r = 0.34
Left-hand mass	r = 0.27	r = 0.31	r = 0.36
Right-leg mass	r = 0.18	r = 0.11	r = 0.27
Left-leg mass	r = 0.18	r = 0.12	r = 0.26

**Table 3 sports-12-00096-t003:** The relationship between artistic swimming elements and hand grip strength (*n* = 17).

	Right-Hand Grip Mean (26.52 ± 4.36 Kg)	Left-Hand Grip Mean (24.29 ± 3.36 Kg)	Hand Grip Mean (25.40 ± 3.73 Kg)
Barracuda scores (6.10 ± 0.64)	r = 0.58 *	r = 0.59 *	r = 0.62 *
Vertical position scores (5.82 ± 0.49)	r = 0.47	r = 0.61 *	r = 0.59 *
Hand grip mean	r = 0.96 *	r = 0.94 *	

## Data Availability

Data is unavailable due to ethical restrictions, but data can be made available upon reasonable request and ethical approval.

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
