# Peer review of "Investigation of Factors Related to Sport-Specific Compulsory Element Execution in Artistic Swimming"

_sports, 2024, doi:10.3390/sports12040096_

Round 1
Reviewer 1 Report
Comments and Suggestions for Authors
Dear Editor,
Thank you for the opportunity to review the manuscript titled "Investigation of factors related to sport-specific compulsory element execution in artistic swimming" (sports-2902174) submitted to the journal Sports. This manuscript investigates the relationship between various factors, including body composition, hand grip strength, lower limb strength, and swimming performance, with the execution of compulsory elements in artistic swimming among young female athletes.
Overall Evaluation:
The manuscript addresses an important topic in the field of artistic swimming, as it explores the factors influencing the performance of compulsory elements. The authors have conducted a comprehensive study involving various assessments and measurements. The findings provide valuable insights into the importance of strength, swimming skills, and the potential benefits of incorporating swim training into the training regimen of young artistic swimmers.
General Comments:
1. The introduction section provides a good overview of the topic and the rationale for the study. However, it could benefit from a more focused literature review, highlighting the specific gaps in the existing literature that this study aims to address.
2. The methods section is generally well-described, but some additional details on the participant selection criteria, training protocols, and statistical analysis methods would strengthen this section.
3. The results section is clearly presented, and the authors have included relevant tables and figures to support their findings. However, some of the results could be more concisely described in the text.
4. The discussion section provides a good interpretation of the findings and their practical implications. However, it could be further improved by addressing the limitations of the study and suggesting future research directions more explicitly.
5. The conclusions section effectively summarizes the main findings and their significance, but it could be more concise.
6. The overall writing and structure of the manuscript are generally clear, but there are some instances of repetition and minor grammatical errors that should be addressed.
Specific Comments:
Page 2, Line 13: Please provide a brief explanation or reference for the statement "Aesthetics are very important, but body composition is equally relevant."
Page 3, Line 53: The authors mention "breaking out of the pre-position" when describing the barracuda technique. It would be helpful to provide a brief explanation or reference for this term.
Page 4, Line 91: The authors state "The effect of breaststroke leg swim training on artistic performance (Results Section 3.2.1) was tested in 38 elite artistic swimmers (Tier 4: International Level)." However, the details of this group are not provided in the Methods section. Please include relevant information about this group, such as age, training experience, and selection criteria.
Page 6, Table 1: Please specify the units for the variables in the table.
Page 8, Line 224: The authors mention "Correlations found between the force plate and the execution of synchro elements show the importance of complex training strategies in this sport." However, the results related to the force plate measurements are not clearly presented in the Results section. Please provide more details on these results or consider rephrasing this statement.
Page 9, Line 245: The authors state "The results of this study demonstrate the relationship between SP and the AS element scores of young artistic swimmers." However, the relationship between swimming performance and barracuda scores is not clearly described in the Results section. Please provide more details or clarify this statement.
Page 11, Line 312: The authors mention "Collecting, analyzing, and correlating performance data of AS athletes, such as strength, flexibility, and body composition, could help trainers and practitioners to create personalized training sessions for young athletes according to age groups and avoid or reduce the possibilities of injuries or fallouts in water sports." While this is a valid point, it would be beneficial to provide some specific recommendations or guidelines for trainers and practitioners based on the findings of this study.
Overall, the manuscript presents valuable research findings and contributes to the understanding of factors influencing artistic swimming performance. By addressing the general and specific comments provided, the authors can further improve the clarity, depth, and impact of their work.
Reviewer 2 Report
Comments and Suggestions for Authors
First of all, the reviewer would like to thank the authors for their work and efforts in trying to improve sports science knowledge.
I consider it an interesting paper to be published in this journal. However, I would like to point out some issues that should be improved
Methods:
- Why did they choose this age? Can they justify with any study the importance of this stage in artistic swimming? What category of competition is at this age?
- The authors should detail the sample a little more, as well as the exclusion criteria.
We thank the authors for the amount of information they analyze in this work. However, their interpretation is confusing. For example, they analyze aspects of body composition at the beginning and end of the season and fire correlate it with performance in the barracuda, vertical or body boost, but it is not clear whether they do so with the measure at the beginning or end of the season.
Confusion is also created by their analysis of the correlation between breaststroke feet with the technical elements of artistic swimming. I think this is not the objective of this article and creates confusion.
Doubts also arise as to when the strength measurements are taken.
The authors should make an effort to clarify all the temporal issues in taking measurements and explain which moments are related.
Results
- The significance values and the symbols used in the tables should be explained at the foot of the table.
- In addition, the classification of strong or low correlation should be detailed in the methods section.
- Table 3 shows the correlation between elements and hand strength. I think there is an error somewhere because the values in the table do not correspond to the interpretation in the following paragraph.
- 3.2.2. Three-Months-Long Breaststroke Leg Swim Training. The design of this study is not understood. Up to this point, the existence of a control group was not contemplated. I think this should be part of another article.
I congratulate the authors for the work developed, and I suggest that they keep in mind other aspects that may have affected the performance in this sport and the results obtained in this research, such as the experience of the swimmers, the hours of accumulated water. The sensitivity and skill achieved over the years and hours of practice may be a factor that may be a supplement to the more physical aspects in this sport and perhaps the significant results found are not so significant with a more experienced sample.
However, the authors make it clear that the results are obtained in a small age sample and therefore not transferable, in absolute terms, to all ages or to the elite of this sport.
Reviewer 3 Report
Comments and Suggestions for Authors
As a first suggestion, I would recommend having a native English speaker proofread the manuscript.
Abstract
The abstract should have a conclusion. Also, the correlations should be explained. Negative or positive correlations should be specified. For example, a correlation was found between mean hand grip strength and barracuda scores as well as between vertical position scores. Is that a positive significant correlation (strong, moderate, weak)?
Introduction
Last paragraph of the introduction: “therefore it is necessary to study them as wide and complex, as possible”. Consider rewriting that sentence. The purpose and hypothesis of the study should be added at the end of the introduction part.
Childhood is the time that extends from infancy to adolescence. You need to be more specific here (last paragraph of the introduction).
Methods
Participants: Body composition instead of body compositions.
Body composition: You need to be more specific as to which instructions your participants followed before the body composition testing, etc.
Force plate testing: How many jumps did they perform? Only one? Did you measure ground reaction force only?
The order of the testing needs to be explained, as well as the warm-up. Did they do all the testing on the same day?
Results
Is Table 2 presenting correlations or p values? This Table is confusing.
The reporting of the results is confusing (24.29 ± 3.36; p = 0.001; t = 3.84). Try to include degrees of freedom and use the guidelines for reporting the statistical values.
Is Table 3 reporting p values?
What does it mean that * means correlation?
Comments on the Quality of English LanguageI believe that an English speaker should proofread this work before further review.
Round 2
Reviewer 3 Report
Comments and Suggestions for Authors
I am happy with the revisions and have no further recommendations or comments.